# Study on the Association between *LRRC8B* Gene InDel and Sheep Body Conformation Traits

**DOI:** 10.3390/genes14020356

**Published:** 2023-01-30

**Authors:** Jiaqiang Zhang, Zhansaya Toremurat, Yilin Liang, Jie Cheng, Zhenzhen Sun, Yangming Huang, Junxia Liu, BUREN Chaogetu, Gang Ren, Hong Chen

**Affiliations:** 1Key Laboratory of Animal Genetics, Breeding and Reproduction of Shaanxi Province, College of Animal Science and Technology, Northwest A&F University, Xianyang 712100, China; 2College of Life Sciences, Northwest A&F University, Xianyang 712100, China; 3Animal Disease Control Center of Haixi Mongolian and Tibetan Autonomous Prefecture, Delingha 817000, China; 4College of Animal Science, Xinjiang Agricultural University, Urumqi 830052, China

**Keywords:** *LRRC8B* gene, InDel, body conformation traits, sheep

## Abstract

Marker-assisted selection is an important method for livestock breeding. In recent years, this technology has been gradually applied to livestock breeding to improve the body conformation traits. In this study, the *LRRC8B* (Leucine Rich Repeat Containing 8 VRAC Subunit B) gene was selected to evaluate the association between its genetic variations and the body conformation traits in two native sheep breeds in China. Four body conformation traits, including withers height, body length, chest circumference, and body weight, were collected from 269 Chaka sheep. We also collected the body length, chest width, withers height, chest depth, chest circumference, cannon bone circumference, and height at hip cross of 149 Small-Tailed Han sheep. Two different genotypes, ID and DD, were detected in all sheep. Our data showed that the polymorphism of the *LRRC8B* gene was significantly associated with chest depth (*p* < 0.05) in Small-Tailed Han sheep, and it is greater in sheep with DD than those with ID. In conclusion, our data suggested that the *LRRC8B* gene could serve as a candidate gene for marker-assisted selection in Small-Tailed Han sheep.

## 1. Introduction

The *LRRC8* gene family was discovered in recent years, and it is composed of five subunits, namely *LRRC8A*, *LRRC8B*, *LRRC8C*, *LRRC8D*, and *LRRC8E* [1,2]. Specifically, the LRRC8 protein is a type of transmembrane protein that exists in various cell membranes in the mammalian body and participates in the formation of the cell VRAC (volume regulation anionic channels) [2,3,4,5,6]. It also has a large influence on the field of cell volume regulation, cell division and migration, apoptosis, cancer drug resistance, and inflammation [7,8,9,10,11,12,13]. However, the LRRC8 protein has not been found in invertebrates, so it may originate in chordates [14] and is considered to be a specific factor in vertebrates [1]. Among the five family members, *LRRC8A* is the most important subunit of the VRAC channel, and it co-expresses with one or more of the other subunits to constitute correct VRAC liveness [2,15]. Therefore, *LRRC8A* is essential for the formation of the VRAC channel. Notably, gene expression should remain in a certain level, and the overexpression of the *LRRC8A* gene may interfere with the normal activity and function of the VRAC channel [15,16].

In addition, studies have shown that the co-expression of *LRRC8A* and other LRRC8 proteins in the *LRRC8* gene family can produce VRACs with different functional characteristics. Moreover, the difference in VRAC activity that the *LRRC8A/B* isomer causes may depend on the cell type and different physiological conditions [17]. The *LRRC8A/D* isomer is the main pathway through which cells are responsible for excreting GABA (γ-aminobutyric acid), taurine, and inositol [4,18]. The *LRRC8A/E* isomer is activated by intracellular oxidation, whereas the *LRRC8A/C* and *LRRC8A/D* isomers are restrained by oxidation [19]. Compared with the *LRRC8A/C* and *LRRC8A/D* isomers, the activity of the *LRRC8A/E* isomer is significantly inhibited by positive membrane potential [16].

The *LRRC8B* gene, a protein coding gene, is an important member of the *LRRC8* gene family. Specifically, the *LRRC8B* protein, a subunit of the channel complex, takes part in regulating the substrate specificity of the channel and participates in intracellular Ca^2+^ homeostasis through an endoplasmic reticulum leakage channel [4,20]. Furthermore, research has found that the protein mediates drug-induced cell death signaling pathways and promotes apoptosis related to anticancer drugs such as cisplatin AVD (apoptotic volume decrease) [21,22,23]. It also plays a variety of roles in the pathology of Alzheimer’s disease and is associated with the progression of the break stage thereof [24]. In addition, the expression of the *LRRC8B* gene is closely related to the lung transplantation process and can be used as a new therapeutic target for lung transplantation complications [25]. However, update there is no report about the function of *LRRC8B* gene in livestock and poultry breeding.

After years of continuous development and improvement, molecular marker breeding with InDel (Insertion and Deletion) technology has become one of the most important methods for livestock and poultry breeding [26]. Recent studies have found that genomic structural variation is very important for the study of population polymorphism, disease susceptibility, and animal phenotype. CNV (copy number variation), InDel, and SNP (single nucleotide polymorphism) are called the three major genomic structural variations. CNV refers to genomic structural variation with length greater than 50 bp, including fragment insertion, deletion, inversion, translocation, duplication, and other types [27]. It is characterized by high resolution, wide genomic coverage, and stable genetic performance [28,29]. It can be detected by chip methods (genomic hybridization chip and SNP chip), sequencing methods (whole genome sequencing and single molecule length sequencing), PCR, fluorescence in situ hybridization (FISH), and other methods [30]. Each of these technologies has its advantages. The one with the highest resolution and widest coverage is whole genome sequencing. However, it is also the most expensive. InDel refers to genomic structural variation with length less than 50 bp, including insertion and deletion of fragments. It is characterized by simple operation, high sensitivity, reliable results, and good stability, and it can be detected by high-resolution melting (HRM), non-denaturing gel capillary (NDGC), amplification refractory mutation system (ARMS), Sanger sequencing, and agarose gel electrophoresis [31]. Among them, Sanger sequencing and agarose gel electrophoresis have been widely used in the detection of candidate genes for economic traits of livestock and achieved brilliant results. The process of this method is as follows: first, primers are designed upstream and downstream of mutation sites, and the specificity and sensitivity of primers are guaranteed. Secondly, PCR was used for amplification. Finally, the target fragment was detected by electrophoresis or sequencing analysis to select the optimal character genotype. SNP refers to single nucleotide polymorphism, which refers to the variation of the position of a certain base, including base insertion, deletion, conversion (purine replaced by purine or pyrimidine replaced by pyrimidine), and transmutation (purine replaced by pyrimidine or pyrimidine replaced by purine). It is characterized by wide distribution, high density, and genetic stability. It can be detected by capillary electrophoresis, gene chip technology, restriction fragment length polymorphism (RFLP), allele specific PCR (AS-PCR), direct sequencing, and other methods [32]. Among them, capillary electrophoresis and gene chip technology are suitable for the detection of unknown SNP mutation; the detection cost is high, and the difficulty is greater. RFLP, AS-PCR, and direct sequencing are suitable for the detection of known SNP mutations, with high detection efficiency.

In terms of the formation mechanism, CNV, InDel, and SNP have different emphases. The formation of CNV is mainly due to non-allelic homologous recombinationg (NAHR), non-homologous end joining (NHEJ), and fork stalling and template switching (FoSTeS). NAHR usually occurs during meiosis, when the copy number is changed due to the exchange of genome sequences. NHEJ is a DNA damage repair mechanism existing in the body, which can reconnect broken DNA and inevitably introduce copy number, resulting in mutation. FoSTeS is the result of DNA replication using the wrong replication template, which also causes mutations to occur. SNP is mainly the variation produced by adaptation to the environment. In the process of biological evolution, mutation is inevitable, and the most common mutation type is SNP. In order to survive, favorable variation is preserved. The generation of InDel is mainly related to the characteristics of genome and DNA replication errors. When the genome sequence is AT(A/C)(AC)GCC and TACCRC, the probability of insertion will be increased. However, sequences containing TATCGC and GCGG are not susceptible to insertion and deletion mutations. InDel is more likely to be produced in regions with high replication; for example, the occurrence frequency of euchromosomes is lower than that of sex chromosomes [33]. In addition, various factors such as transposon replication and insertion, mobile element insertion, abnormal sequence recombination, and unequal exchange of similar duplicate copies can also cause InDel [34].

As germplasm resources are dwindling, it is crucial to breed high-quality sheep. There are many types of sheep breeds in China, but most of them have poor performance and cannot satisfy people’s increased demand. Chaka sheep and Small-Tailed Han sheep are two good breeds in our country, widely distributed throughout the country. Chaka sheep, which live in Wulan County, Qinghai Province, are a famous hybrid breed of sheep used for local wool and meat with high meat quality but low yield [35]. On the other hand, Small-Tailed Han sheep are widely distributed and have strong adaptability and are resistant to rough feeding. It is a famous sheep breed for both meat and wool in China and globally, and it has been named as the “national treasure” of China [36]. However, the genetic improvement of the body conformation traits of local Small-Tailed Han sheep in China is still urgently needed.

## 2. Materials and Methods

### 2.1. Experimental Materials

A total of 269 healthy Chaka sheep aged from 1 year old to adults were randomly selected from Chaka Town, Wulan County, and they had no blood relationship with each other. This included 83 male sheep, 101 female sheep, and 85 wether sheep for which experimental records were made. A total of 149 healthy Small-Tailed Han sheep with no blood relations were also randomly selected from Ruilin Technology Breeding Co., Ltd., Yongjing County, Gansu Province, China, including 85 male sheep and 64 female sheep for which experimental records were made. Notably, these sheep in each growth stage were under the same feeding management and nutrition conditions.

### 2.2. Sample Collection and Measurement of Body Size

Blood samples were collected from all experimental animals, following which they were fully mixed with anticoagulant and stored at −80 °C. Simultaneously, the body size traits of the experimental animals were recorded, including the wither height, body length, chest circumference, and body weight of the Chaka sheep, as well as the body length, chest width, wither height, chest depth, chest circumference, cannon bone circumference, and height at the hip cross of the Small-Tailed Han sheep. To reduce random errors in this experiment, the same person measured and recorded the same trait.

### 2.3. Primer Design

The sheep’s *LRRC8B* gene sequence (serial number: NC_056054.1), which the NCBI database published, was referred to, and gene-specific primers were designed using Primer. The specific primers were submitted to Sangon Bioengineering Co., Ltd (Shanghai, China) for synthesis (Table 1).

### 2.4. The PCR Amplification System and Procedures

A 10 μL PCR amplification system was used, including 5 μL of PCR Master Mix, 0.5 μL of sheep DNA, 3.9 μL of ddH_2_O, and 0.3 μL of upstream and downstream primers each. The procedure used was as follows: pre-denaturation for 5 min at 95 °C; denaturation for 30 s at 94 °C, as well as at 68–50 °C cycle of 30 s, with each cycle decreasing by 1 °C, and a 72 °C extension for 30 s for a total of 18 cycles; denaturation for 30 s at 94 °C, annealing for 30 s at 50 °C, and an extension for 30 s at 72 °C for a total of 26 cycles; an extension for 10 min at 72 °C, following which it was stored at 12 °C. Then, the classification of the reaction products was detected using 3.5% agarose gel electrophoresis.

### 2.5. TA Cloning Experiment

The first step was to select the DNA of 25 heterozygous genotypes and mix them together. In the second step, the same system and procedure were used for PCR amplification. The third step was to verify the PCR products by agarose gel electrophoresis. The fourth step was to purify the amplified product. Step five was to use Tsingke Biotechnology Co., Ltd. (Beijing, China) The 5xpClone007 Versatile Simple Vector Mix was developed to construct the T Vector (junction site: CCCTT AAGGG). The sixth step was the transformation of connection products. The seventh step was to inoculate the transformed product into a bacterial petri dish. Step eight was to select monoclones, and second generation sequencing was used for verification.

### 2.6. Data Collation and Analysis

SPSS23.0 software was used to conduct association analysis between the body size data of the Chaka and Small-Tailed Han sheep populations and the InDel typing results. We used the website SHEsis (http://analysis.bio-x.cn (accessed on 10 March 2021)) to analyze the HWec (Hardy–Weinberg equilibrium constant), Ho (homozygosity), He (heterozygosity), PIC (polymorphism information content), genotype frequency, and allele frequency of *LRRC8B* gene polymorphisms in the sheep population. All data were presented as means ± SE [37]. A conventional linear model was also used to explore the effects of the elements on sheep traits: y_mn_ = μ + G_m_ + e_n_, where μ was the average population value, G_m_ was the genotype fixed effect, and e_n_ was the stochastic error.

## 3. Results

### 3.1. Screening of the InDel Locus of the LRRC8B Gene

The InDel locus was located in the exon region of the *LRRC8B* gene on chromosome 1. After PCR amplification, it was found that there was a 4 bp InDel mutation in this gene, which was located at 74,215–74,218 bp of this gene and 66,963,967–66,963,970 bp on chromosome 1 (Figure 1).

### 3.2. PCR Results and Genotyping

The 4 bp mutation site of the *LRRC8B* gene was amplified using PCR according to the specific synthesized primers, and the genotyping of the *LRRC8B* gene was analyzed using 3.5% agarose gel electrophoresis. The results showed that *LRRC8B* gene mutation sites include two genotypes: heterozygous (ID) and homozygous deletion (DD), and the DNA sequence is consistent with the electrophoretic figure (Figure 2 and Figure 3).

### 3.3. InDel Genetic Parameter Analysis of the LRRC8B Gene

The results of the genotyping were statistically analyzed to calculate the HWec, Ho, He, PIC, genotype frequency, allele frequency, etc., which are presented in a table (Table 2). In the Chaka sheep, the frequency of the D gene was 0.714, so it belonged to the dominant gene but not in the HWec (*p* < 0.05). Furthermore, He was 0.409 and PIC was 0.409, showing moderate polymorphism. In the Small-Tailed Han sheep, the frequency of the D gene was 0.795, so it belonged to the dominant gene in the HWec (*p* < 0.05), whereas He was 0.326 and PIC was 0.326, so it belonged to moderate polymorphism.

### 3.4. Association Analysis between LRRC8B Gene Polymorphism and Body Conformation Traits

In the 269 Chaka sheep, wither height, body length, chest circumference, and body weight were measured and underwent associated analysis with the different genotypes of the *LRRC8B* gene. However, no obvious associations were found among these four traits (Table 3). Moreover, seven body conformation traits, including body length, chest width, wither height, chest depth, chest circumference, cannon bone circumference, and height at the hip cross, of the 149 Small-Tailed Han sheep were measured, and association analysis was conducted with the different genotypes of the *LRRC8B* gene. The results showed that there was a significant association with chest depth (*p* < 0.05) with a *p* value of 0.044. Additionally, the DD genotype of the Small-Tailed Han sheep was better than their ID genotype. However, no significant association was found with the other six body conformation traits (Table 4).

## 4. Discussion

In this study, a tiny InDel mutation of the sheep *LRRC8B* gene was screened, which was only 4 bp. Using ordinary agarose gel electrophoresis, however, the ID and DD genotypes could not be distinguished. Nevertheless, the electrophoresis provided results for the following two items: a stripe for mutations with a length of 342 bp and 338 bp (indistinguishable), with the other presumably being a heterologous double strand [38].

Heterologous double strands are formed from the complementary pairing of the single-stranded bases of the double-stranded molecules of both parents. However, due to the different sources of the two single DNA strands, the bases are not completely paired, and it is easy to form an unpaired ring structure. Therefore, to verify the heterologous double-stranded structure of this experiment, a TA cloning experiment was conducted. The sequencing results showed not only ID and DD genotypes but also many sequences that were significantly different from the PCR products, which confirmed our speculation about heterologous double chains.

Our results showed that there was no significant association between the InDel of *LRRC8B* gene and body conformation traits (wither height, body length, chest circumference, and body weight) in the 269 Chaka sheep population. Surprisingly, the ID genotype had better traits than the DD genotype, with the best association being for body length, followed by body weight and wither height, and the worst association being for chest circumference. We speculated that individuals with ID genotype could regulate the expression of *LRRC8B* or other genes, thus promoting the growth and development of Chaka sheep. Among the 149 Small-Tailed Han sheep population, the InDel of *LRRC8B* gene was significantly correlated with the chest depth of Small-Tailed Han sheep, and the association coefficients reached 0.044. The traits of DD genotype were superior to that of ID genotype. However, InDel of *LRRC8B* gene did not show significant association with other six traits (body length, chest width, wither height, chest circumference, cannon bone circumference, and height at hip cross). The dominant traits of ID genotype individuals included body length, wither height, cannon bone circumference, and height at hip cross, whereas the dominant traits of DD genotype individuals included chest width and chest circumference. The best association was cannon bone circumference, and the worst was body length. The HWec of Chaka sheep and Small-Tailed Han sheep were both lower than 0.05, indicating that the gene frequency and genotype frequency of the two sheep breeds were in a state of balance. The population genetic diversity showed a positive association with polymorphism information content. The PIC and He of Chaka sheep and Small-Tailed Han sheep were both greater than 0.25 and less than 0.5, which belonged to the moderate polymorphism level. The genetic variation was rich; we could increase the mining of beneficial mutations and promote the breeding process of Chaka sheep and Small-Tailed Han sheep.

In the field of livestock and poultry breeding, InDel molecular marker technology has made remarkable achievements, one example of which shows that the InDel mutations of *PLAG1* and *SIRT4* genes are significantly correlated with the body conformation traits of several cattle breeds [31,39]. In addition, the InDel mutations of the *Cry2* and *GDF9* genes have been found to be significantly correlated with the litter size of sheep and goats, respectively [38,40]. Further, the 12 bp length mutation of the *Oct4* gene is associated with the reproductive traits of male piglets [41], whereas the 65 bp mutation of the *GOLGB1* gene is significantly correlated with the growth and carcass traits of chickens [42]. In addition, some research has found that the 10 bp InDel of the *FecB* gene is significantly correlated with the number of offspring produced in Chinese Australian White sheep [43]. Similarly, in our study, the 4 bp InDel of the exon region of *LRRC8B* gene was significantly correlated with the chest depth of Small-Tailed Han sheep, and the DD genotype individuals showed better performance. The insertion–deletion mutation of the *CREB1* gene, which is involved in the regulation of fat metabolism in sheep adipose tissue, is also significantly related to the body conformation traits of sheep [44]. These results indicate that they could be used as molecular markers for sheep breeding. However, these are just a few of the studies that have been conducted.

Chaka sheep have adapted to the environment of Chaka Salt Lake over time, forming a unique gene pool with abundant alleles [35]. Although this experiment did not find a significant association between genotypes and body conformation traits, it did find that the InDel locus of the *LRRC8B* gene is associated with other unchecked traits. Therefore, the InDel molecular marker still has great application prospects for animal breeding. Furthermore, the Small-Tailed Han sheep is an excellent local sheep breed in China with high performance and can be used as an ideal parent candidate in meat production. In this study, it was found that the mutation of the *LRRC8B* locus at 74,215 bp–74,218 bp is significantly associated with chest depth in Small-Tailed Han sheep (*p* < 0.05), and chest depth is closely and positively related to sheep body condition. That is, the deeper the chest, the fatter the sheep. However, further research is needed to explain this.

In order to test the association between the InDel site of *LRRC8B* gene and the chest depth of Small-Tail Han sheep, we plan to construct *LRRC8B* gene interference and overexpression vector and transfect sheep muscle primary cells to detect the growth and development of cells (such as cell proliferation and differentiation). Meanwhile, ESEfinder 3.0 online software was used to predict the splicing point changes of the pre-mRNA of the gene, and Phyre2 software was used to predict the protein structure and function of the gene.

To date, there have been few functional studies on the *LRRC8B* gene, most of which have been in the field of medical research and are widely used in cancer treatment [19]. In addition, the direction of research has been toward the function of related channel structures, although the structure of genes themselves and the function of other DNA regions remain unclear. Therefore, starting from the field of livestock and poultry breeding, in this study, InDel molecular marker technology was used to explore the association between the InDel of the *LRRC8B* gene and the body conformation traits of Chaka and Small-Tailed Han sheep.

## 5. Conclusions

In conclusion, this study found a new InDel of the *LRRC8B* gene in the Small-Tailed Han sheep population and suggested that it can be used as a molecular marker to improve meat yield.

## Figures and Tables

**Figure 1 genes-14-00356-f001:**
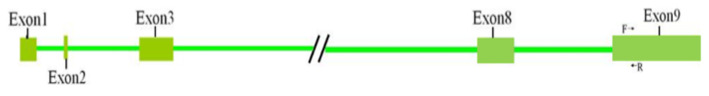
*LRRC8B* gene structure. Note: the green boxes in the figure indicate exons, and “F” and “R” refer to the upstream and downstream primers, respectively.

**Figure 2 genes-14-00356-f002:**
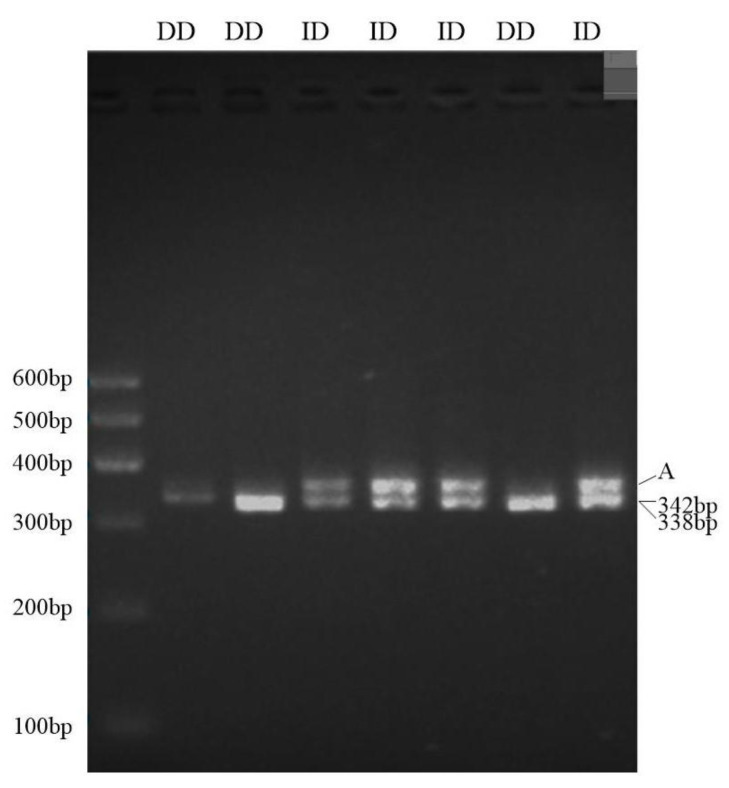
The *LRRC8B* gene InDel site using gel electrophoresis. Note: the letter “A” indicates a heterogenous double chain.

**Figure 3 genes-14-00356-f003:**
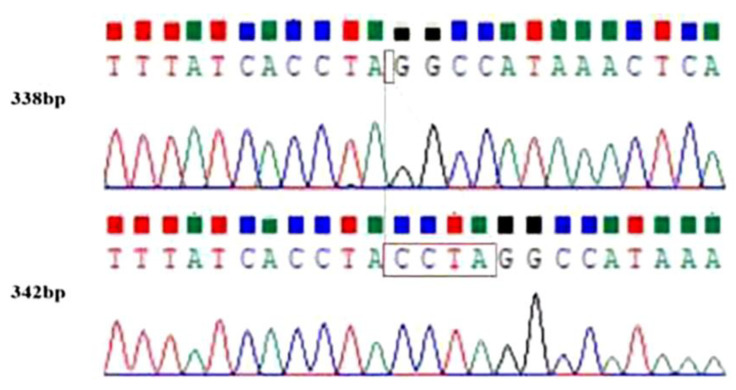
The *LRRC8B* gene InDel site via sequencing. Note: the red line and box show the location of the base mutation.

**Table 1 genes-14-00356-t001:** Primer information.

Gene		Primer Sequence (5′ to 3′)	Amplification Length/bp
*LRRC8B*	Forward primer	ACTTGGAGGGTAGAATGGGG	342 bp
	Reverse primer	ACAGGCAGGCACTTTCTCAG	

**Table 2 genes-14-00356-t002:** Analysis of the InDel genetic parameters of the *LRRC8B* gene in Chaka sheep and Small-Tailed Han sheep.

	Number	Genotype Frequencies		Gene Frequencies		HWec	Genetic Parameter Estimation			
		ID	DD	I	D		Ho	He	Ne	PIC
CKS	269	0.572 (154)	0.428 (115)	0.286	0.714	*p* < 0.05	0.591	0.409	1.691	0.409
STHS	149	0.409 (61)	0.591 (88)	0.205	0.795	*p* < 0.05	0.674	0.326	1.483	0.326

**Table 3 genes-14-00356-t003:** Association analysis of *LRRC8B* gene polymorphism and body conformation traits in Chaka sheep.

Body Conformation Traits	Genotype (Mean ± SE)		*p* Value
	ID	DD	
Wither height/cm	66.29 ± 4.70	65.57 ± 4.50	0.470
Body length/cm	72.18 ± 6.80	72.14 ± 7.30	0.314
Chest circumference/cm	89.95 ± 8.63	88.63 ± 8.26	0.589
Body weight/cm	54.87 ± 13.42	53.26 ± 13.18	0.367

**Table 4 genes-14-00356-t004:** Association analysis of *LRRC8B* gene polymorphism and body conformation traits in Small-Tailed Han sheep.

Body Conformation Traits	Genotype (Mean ± SE)		*p* Value
	ID	DD	
Body length/cm	59.60 ± 5.63	59.44 ± 6.17	0.946
Chest width/cm	19.28 ± 2.67	19.51 ± 3.53	0.907
Wither height/cm	63.61 ± 4.09	63.36 ± 4.44	0.841
Chest depth/cm	27.23 ± 2.02 ^b^	28.13 ± 2.85 ^a^	0.044 *
Chest circumference/cm	72.11 ± 5.74	72.64 ± 6.47	0.685
Cannon bone circumference/cm	7.24 ± 0.66	7.10 ± 0.79	0.382
Height at hip cross/cm	63.59 ± 3.85	62.87 ± 4.46	0.406

Note: different lowercase letters on the shoulder indicated significant differences (*p* < 0.05), whereas no letters on the shoulder indicated no significant difference (*p* > 0.05). * indicates a significant difference.

## Data Availability

Not applicable.

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
