# Peer review of "Study on the Association between LRRC8B Gene InDel and Sheep Body Conformation Traits"

_genes, 2023, doi:10.3390/genes14020356_

Round 1

Reviewer 1 Report

The manuscript is interesting and presents relevant information regarding the possible effects of the LRRC8 gene on sheep growth traits.

I was surprised that in the introduction, the authors already presented their results (L79-82). This is not common and I suggest removing it.

In the title of the manuscript, it is stated that the relationship between the gene and an index of growth traits will be studied. But, nowhere in the text is this index. It should be noted that the analysis of traits separately should not be referred to as an index. Index is a single value of the union of characteristics weighted by a weight, which can be percentage or economic. There are no indexes in the manuscript. I recommend changing from the title to the text, removing the reference to the growth traits index.

Another point that I suggest attention is that when talking about an association study, statistically a correlation analysis or a regression analysis is expected. From the description of the material and methods, and from the presentation of the results in tables 3 and 4, an ANOVA was performed, with comparison of means (it was not indicated which test was used for the comparison). Therefore, I suggest a resubmission of the manuscript considering this issue. Either reanalyze as a correlation study or change the project scope to performance differences according to animal genotype.

Finally, I suggest caution and parsimony regarding the presentation of results and conclusions. The sample used used animals aged one year or older. If the authors wish to study the association of the gene with the growth of animals, it would be more appropriate to use younger animals, in a more active growth phase. The growth curve studies in sheep clearly show that after one year the intensity of growth greatly reduces, tending to stability.

Author Response

Thank you for your useful comments and suggestions on the language and the structure of our manuscript. According to your recommendation, we have modified the manuscript accordingly, and the detailed corrections are listed below point by point.

Reviewer 2 Report

In this manuscript, the authors attempted to explore the relationship between the LRRC8B variations and the growth traits in two native sheep in China. The study found a new InDel of the LRRC8B gene in the Small-tailed Han sheep population and suggested that it can be used as a molecular marker to improve meat yield.

Overall, this manuscript is a well-written, interesting, and valuable result. The following are some of the minor issues.

Point1:  In lines 125 to 127 and 131. No references have been cited.

Point2: Could you please, mention the descriptions of the abbreviation for each Table (2, 3, and 4)?

Pint3: Please try to compare your results with different studies, especially in the discussion parts.

Author Response

(The authors gave the same response as above.)

Reviewer 3 Report

The manuscript tested the association of the LRRC8B with body conformation traits in sheep. The sample size is relatively small. I have some concerns regarding the current approach for the identification of mutation/Idel.

Line 15: MAS is not a hot topic; it does not have many applications in livestock.

Line 15-16: How many variations did the authors check?

The traits are body conformation traits, not growth traits.

Line 22: What is the position of this mutation in the gene?

Line 25: For which traits?

Line 85-92: How did the authors check the relationship between these animals, are there any pedigree information available for them?

What are the ages of animals (in days) since body conformation traits are age-related, at least the authors should consider age as co-variate or effects on the models.

Line 115: Did the authors purify the PCR product before sequencing?

Line 117-118: How did the authors find the Idel, by sequencing?

Line 132: What did the authors mean by “stochastic error”, uncapitalized e should be used in the model.

Line 140: Without sequencing all samples, I do not believe the authors could be able to separate the bands based on gel electrophoresis.

Tables 3 and 4: Did the authors check the percentage of variance explained by this mutation or Idel?

Author Response

(The authors gave the same response as above.)

Round 2

Reviewer 1 Report

The edits and modifications made to the text seem to me to be sufficient, so that the suggestions were met.

Author Response

Thank you for all your efforts. We have made corresponding modifications according to your requirements

Reviewer 3 Report

The authors only focus on editing the comments in English; however, the quality of the writing is still very poor. There is no improvement in the scientific content in the revised version.

Line 19: Change such as to including (if the authors list all the traits)

Line 23: these sheep: did the authors mean both breeds?

The abstract should contain a sentence about the method of genotyping and statistical analyses

Line 24: Change “gene correlated” to the gene was significantly associated with …

Line 26: Marker-assisted selection for which traits?

How about the conclusion for Chaka sheep?

Author Response

(The authors gave the same response as above.)
